# Tire Recycled Rubber for More Eco-Sustainable Advanced Cementitious Aggregate

**Matteo Sambucci [1,2], Danilo Marini [2] and Marco Valente [1,2,*]** 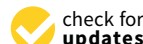

[1]    Department of Chemical and Material Engineering, Sapienza University of Rome, 000184 Rome, Italy; matteo.sambucci@uniroma1.it

[2]    INSTM Reference Laboratory for Engineering of Surface Treatments, Department of Chemical and Material Engineering, Sapienza University of Rome, 000184 Rome, Italy; danilo.marini@uniroma1.it

*    Correspondence: marco.valente@uniroma1.it; Tel.: +39-06-4458-5582

**Abstract:** This research focused on using ground tire rubber (GTR) with different grain sizes as a replacement for the mineral aggregates used in a cement-based mixture suitable for extrusion-based Additive Manufacturing. The use of two types of GTR particles and the possibility to apply rubberized mixtures in advanced manufacturing technologies are the innovative aspects of this work. At the base of this strategy is the possibility of achieving cementitious aggregates, which would potentially be improved regarding some technological-engineering requirements (lightness, thermal-acoustic insulation, energy dissipation capacity, durability) and environmentally sustainable. The integration of waste tires into cement-based materials is a promising solution for the reuse and recycling of such industrial waste. In addition, this approach may involve a considerable reduction in the use of natural resources (sand, water, coarse mineral aggregates) needed for the building materials production. The purpose of the research was to investigate the effect of sand-GTR replacement on certain chemical-physical properties of mixtures (permeable porosity, surface wetness, and water sorptivity), closely related to material durability. Besides, the role of rubber on the printability properties of the fresh material was evaluated. GTR fillers do not alter the rheological properties of the cement material, which was properly extruded with better print quality than the reference mixture. Concerning chemical-physical characterization, the GTR powder-granules synergy promotes good compaction of the mixture, hinders the cracks propagation in the cement matrix, decreases the permeable porosity, improves the surface hydrophobicity and preserves optimal water permeability.

**Keywords:** additive manufacturing; cement-rubber mixtures; tire recycling; permeable porosity; wetting properties; water sorptivity; cement material durability

## 1. Introduction

Thanks to the development of Additive Manufacturing (AM) in the construction sector, concrete technology has considerably advanced in engineering, architectural and environmental terms. The main benefits of additive processes, compared to standard manufacturing, are summarized in the following aspects: higher fabrication speed, lower waste materials production, ease of prototyping and high freedom in terms of the design of shapes [1]. The combination of digital fabrication and cement-based materials promoted the development of innovative manufacturing processes for both small-scale production (building components, urban furnishings) and large-scale fabrication (infrastructures, housing modules). A detailed overview of additive technologies with building materials is reported in [2]. Among the technologies currently in operation, the extrusion-based method is the most widespread in academic and industrial sectors [1]. The deposition apparatus is very similar to the common Fused Deposition Modeling (FDM) printers for polymers: the cement-based material is

extruded by a digitally controlled nozzle, through layer-upon-layer deposition, until the pre-designed object is built. In this context, the composition of the material plays a key role in the quality of the final product. The rheology of the cement mixture must be properly studied to obtain the best printability properties, such as extrudability, workability, buildability and inter-layer adhesion. The correct balance of these parameters ensures a fresh material that can maintain its shape during deposition, able to support the successive layers immediately after extrusion and provide the bond between the adjacent layers avoiding the formation of voids and structural defects. These aspects are fundamental to the structural integrity and mechanical strength of the hardened material [3]. The scientific literature highlights several studies related to the optimization of mixtures with metal fillers [4] or reinforcement fibers (carbon, glass, basalt, polypropylene fibers) [5,6] mainly to improve the workability and mechanical strength of the material.

This research work is based on the modification of "printable" cement mortar by the total replacement of fine mineral aggregates with rubber fillers deriving from waste tires. In traditional manufacturing, the use of ground tire rubber (GTR) as aggregate in the cement matrix is a widely used strategy both to improve some physical and engineering properties of the building material (deformability, vibro-acoustic damping, thermal insulation, lightness) and to achieve interesting effects from an environmental point of view (disposal of waste tires and reduction in the use of natural resources) [7]. Generally, researchers in this field focused their attention on incorporating a single type of polymer aggregate (generally, 0.5–5 mm crumb rubber) as a partial substitute for fine mineral aggregate (20%/25% by volume replacement) [8,9].

The innovative aspect of this work is the use of two different GTR particle grains (0–1 mm rubber powder and 2–4 mm rubber granules) in cement compound for additive construction and evaluate how their synergy affected the several chemical-physical performances of the material. For this purpose, two rubberized mixtures have been designed and developed:

- Singly-sized rubber filler mixture (SSM) was obtained for total sand replacement with GTR powder
- Combined-sized rubber filler mixture (CSM) was obtained for total sand replacement with specific GTR granules/powder ratio (25% by volume of rubber powder—75% by volume of rubber granules)

In addition, a reference mortar (Ref.) containing 100% sand was also prepared to compare its properties with those of the rubber-cement samples. Specifically, in the CSM mix production, a larger amount of polymer coarse aggregate was selected than the powder. The fine aggregate encourages material compaction and ensures a better voids-filling ability. Rubber coarse aggregate, on the other hand, performs greater engineering functionality in terms of workability improvements, ductility, micro-cracks reduction, acoustic and thermal inertia [10,11].

The paper describes an experimental program that examines the effect of GTR/sand replacement on the print quality of the cement material, linked to the fresh mix rheology. Later, the research investigated how the two types of GTR fillers affect the chemical-physical performances of the cement compound in terms of morphological properties, surface wetness, and water permeability. These properties are closely related to the durability performances of cement-based material, one of the fundamental requirements for applications in the building and architectural fields.

## 2. Results and Discussion

### 2.1. Print Quality Investigation

The additive manufacturing of the slabs made it possible to evaluate the correct rheology of cement mixtures and the structural characteristics of hardened materials. The presence of GTR fillers, replacing the mineral aggregate, preserves suitable printability properties of the fresh compounds, which were extruded without interruptions or collapse phenomena. Besides, the printed objects showed good surface finish and dimensional conformity with reference to 3D CAD design.

The visual inspection of the internal morphology of the specimens revealed better compaction and uniformity of the rubberized mixtures compared to the Ref. mix (see Figure 1a). The good homogeneity of GTR modified compounds resulted in a defect-free structure while, in the standard mix, it can be seen the presence of inter-layer voids (see Figure 1b). The non-polar character of the rubber additives could changes the overall surface tension of the mixtures (lower rubber-cement additives interfacial interactions) with consequent greater fluidity than the "neat" mixture. This effect results in easier flow that promotes the filling of gaps before hardening and minimizes the formation of voids in the interstices between the filaments. Our hypothesis is in good agreement with the results reported by previous studies [12,13]. For instance, Aiello and Leuzzi [12] observed that the rheological behaviour of fresh concrete changes from "fluid" to "hyper-fluid" when the mineral aggregates were replaced with rubber aggregates.

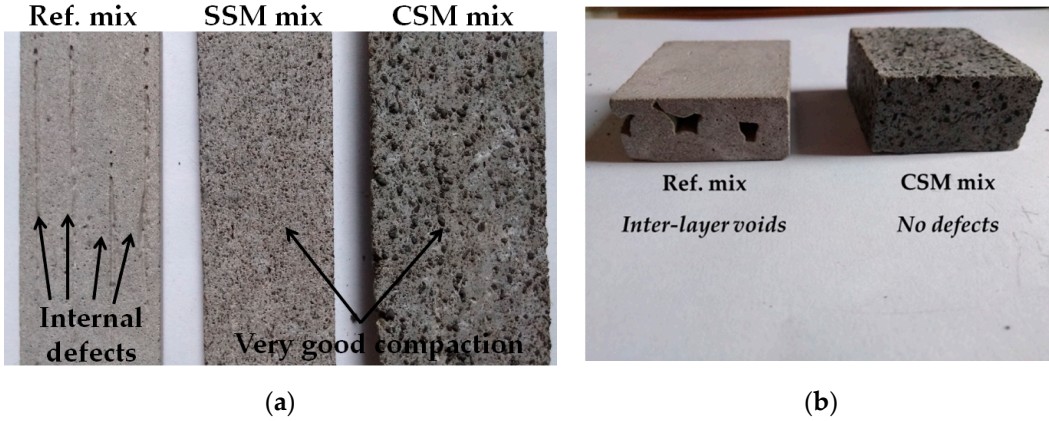

(**a**)          (**b**)

**Figure 1.** Inter-layer adhesion evaluation in cementitious samples (**a**) and investigation of structural defects (**b**).

### 2.2. Porosity Measurements

The results of the vacuum saturation method are shown in Figure 2. Modified mixtures with GTR aggregates exhibited a lower permeable porosity than the standard mix. This behaviour is closely related to three reasons [14]: (a) Ref. sample porosity is high because of the use of porous mineral aggregates; (b) rubberized mixtures were developed with less water than the standard mixture, so during the aging process the formation of air bubbles and voids is less relevant; (c) thanks to their viscoelastic properties, rubber particles more effectively hinder the cracks propagation (due hygrometric shrinkage) in the cement matrix than stiff mineral aggregates.

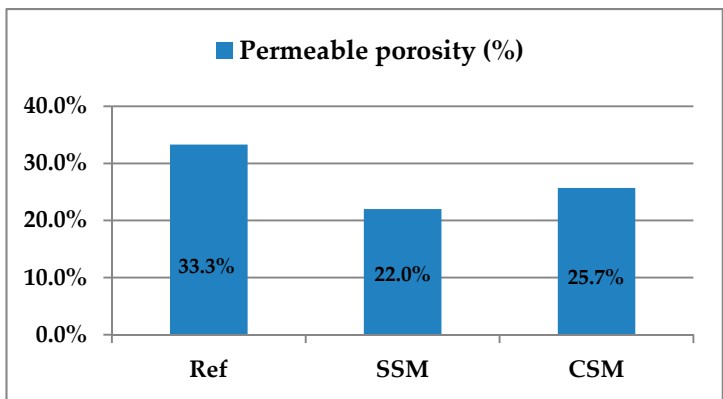

**Figure 2.** Permeable porosity (%) experimental results.

Comparing SSM and CSM mixtures, it is possible to observe that the presence of GTR granules lightly increases the voids percentage than the mixture with only GTR powder. Coarse rubber

aggregates provide better performance on cracking blocking but have poor adherence to the matrix. This effect produces interface voids that affect the open porosity of the material. On the other, powder particles have a larger specific surface area than granules, so they ensure better adhesion with cement resulting in a more compact microstructure [10].

Optical micrographs, acquired with 16× magnification, confirm the trend determined by the experimental permeable porosity results. For microscopy analysis, Ref. sample was compared with CSM sample (containing both GTR fillers). The concentration and size of voids (labeled in yellow) in the rubberized sample (Figure 3b) are much lower than in the "neat" material (Figure 3a).

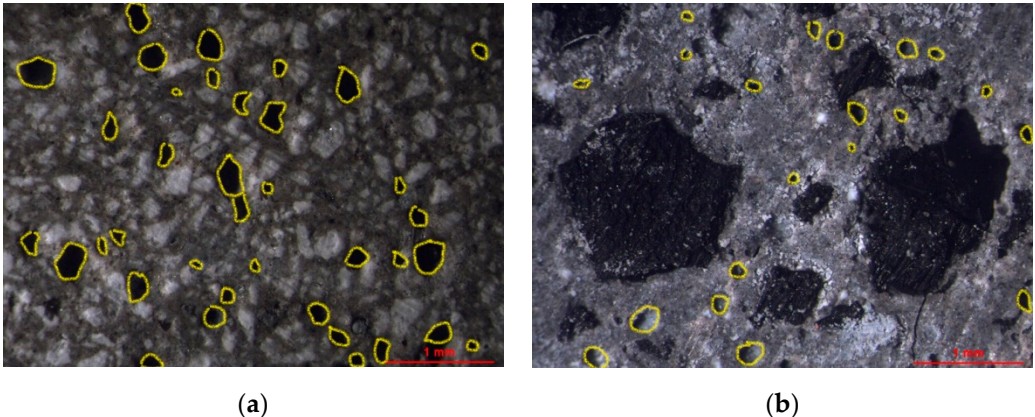

(**a**)                                                    (**b**)

**Figure 3.** Optical microscopy analysis on cement-based samples: porosity distribution in Ref. (**a**) and CSM (**b**).

### 2.3. Water Contact Angles (WCAs) Test

WCAs test results (see Table 1) show that GTR fillers reduce the wettability of the cementitious surface (increase in WCA value).

**Table 1.** Average WCA values for each cement-based mixture.

| Mixture Type | Average WCA |
|:---:|:---:|
| Ref. | 54° ± 15° |
| SSM | 80° ± 14° |
| CSM | 111° ± 38° |

A comparison among the Ref. sample and the rubberized mixtures is reported in Figure 4, with the high-resolution images of the respective surfaces captured during the WCA experiment. SSM and CSM test surfaces (Figure 4b,c, respectively) appear rougher than Ref. surface (Figure 4a) due to the distribution of polymer aggregates incorporated in the cement paste. A very different surface-water droplet interaction is observed on the samples: "neat" cement matrix promotes faster water absorption and greater drop spreads over the surface compared to the rubberized surfaces. The GTR particles act as hydrophobic sites, preventing the deposition of the liquid on the cement matrix and its potential permeation in the material. The size of polymer particles plays a key role in this water-repellent effect. The presence of GTR granules (CSM) decreases the wetting properties, as the probability of water drop settles on the hydrophobic site is higher than the presence of only GTR powder (SSM). The coarse rubber aggregates expose a greater contact surface for the water drop, which tends to remain stable and undeformed after deposition (see Figure 4c). The wetting behavior of the CSM can be classified as hydrophobic (WCA > 90°). Thus, filler hydrophobicity prevails over the absorbent behavior of the hydrophilic cement matrix [15].

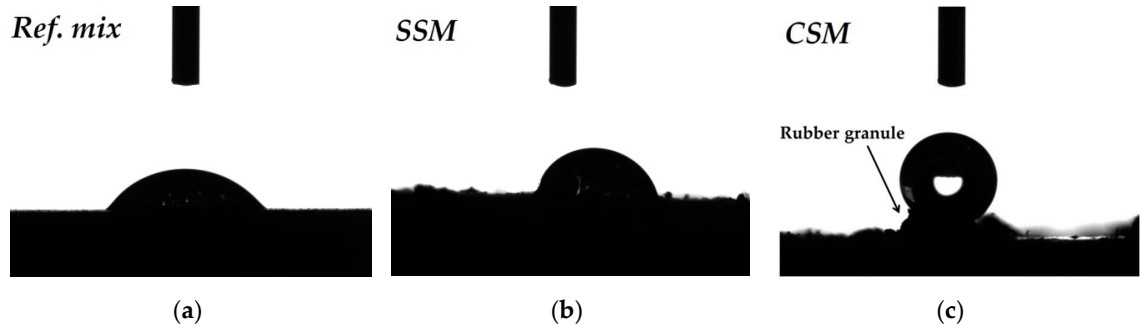

(**a**)            (**b**)            (**c**)

**Figure 4.** Water drop—cement surface interaction: Ref. (**a**), SSM (**b**) and CSM (**c**) samples.

*2.4. Water Sorptivity Test*

For this experimental study, two specimens of each mixture were tested, and the average S-index was determined. The quality of the cement mixtures was evaluated by considering the recommended sorptivity values proposed by Papworth and Grace [16]. These reference sorptivity performance classes are listed in Table 2.

**Table 2.** Sorptivity performance classes proposed by Papworth and Grace.

| Sorptivity (mm/min$^{0.5}$) Performance Classes | | |
|---|---|---|
| **Poor** | **Acceptable** | **Very Good** |
| >0.2 | 0.1 to 0.2 | <0.1 |

The water sorptivity index results are plotted in Figure 5.

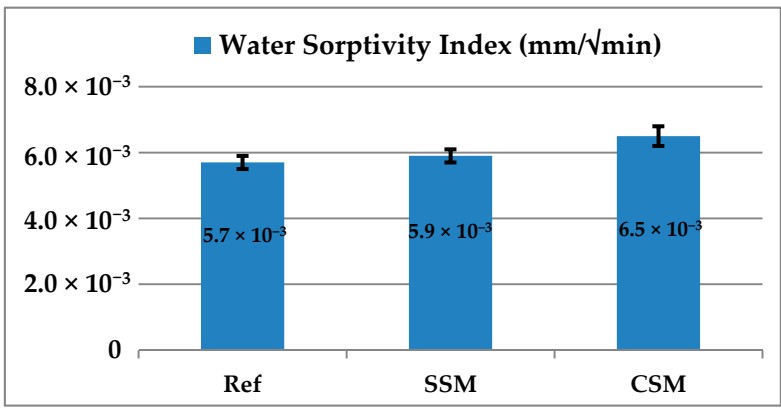

**Figure 5.** Water sorptivity of Ref. mix and rubber-cement mixtures (SSM and CSM).

All cement-based mixtures can be classified as "very good" since the S-index values are less than 0.1 mm/min$^{0.5}$. In rubberized mixtures, a slight increase in sorptivity is observed compared to the standard material. This trend may be related to the weak adhesion between the GTR particles and the cement matrix, which can generate additional access ducts for liquid penetration.

As highlighted by the SEM (MIRA 3 FEG-Scanning Electron Microscope) micrograph in Figure 6a, the weak rubber-cement interface is more relevant in the presence of GTR granules. This effect implies a greater water penetration in the CSM sample than other cement mixtures, confirming the trend of water sorptivity previously analyzed. On the other hand, in the SSM mix, the GTR powder-cementitious matrix adhesion appears more cohesive (Figure 6b), confirming the functionality of the fine fillers to ensure better material compaction and to minimize interface voids [10].

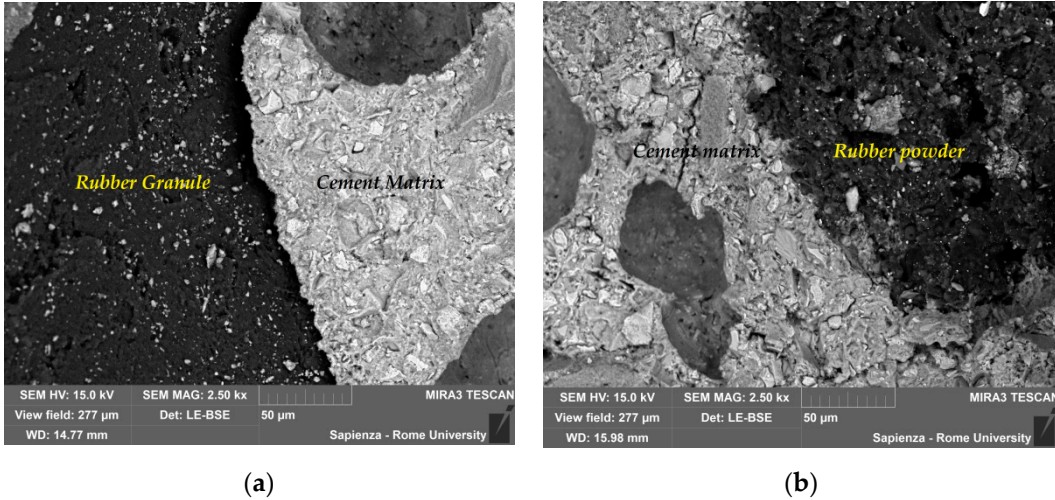

(**a**)                                  (**b**)

**Figure 6.** SEM micrograph of rubber fillers-cement matrix interface: rubber granule–cement adhesion (**a**) and rubber powder-cement adhesion (**b**).

## 3. Materials and Methods

### 3.1. Experimental Extrusion System for Cement-Based Mixtures

The laboratory testing of fresh printing mixtures and the additive manufacturing of the samples were performed at the Mechanical Engineering Department of the Marche Polytechnic University (Ancona, Italy). In general, the extrusion system comprises a COMAU 3-axis robotic-arm equipped with a PVC deposition nozzle (Ø = 10 mm), a pumping system, an aluminum tank with piston, a vibrating platform and a printing surface. The pumping system applies 4 bar pressure to the cylindrical tank containing the fresh cement mortar. This process drives the piston, which promotes the entry of the mixture into the deposition nozzle. The nozzle is connected to the robotic arm that moves according to the instructions imposed by the control software. CURA® control software (Utrecht, Netherlands) elaborates the 3D CAD file of the object to be made and allows to select the manufacturing parameters (printing speed, infill, layer thickness, print path). The 3D design is completed in a layer by layer process through a combined motion of the robotic arm in X, Y, and Z direction.

### 3.2. Materials Preparation

In this work, the rubberized mixtures (labeled as SSM and CSM respectively) were developed from a standard printable cement mortar (labeled as Ref.). Ref. mix (provided by INNOVAcrete®, Ancona, Italy) consists of Type I Portland cement, 0–0.4 mm limestone sand, 0.35 *w/c* ratio and chemical additives (Silica fume, Calcium oxide, superfluidifyng agent and reducer additive) in proper proportions. GTR powder (Figure 7a) and granules (Figure 7b) were provided by ETRA (European Tyre Recycling Association) and were obtained by ambient mechanical grinding of scrap tires.

Fillers size was evaluated by optical microscope analysis (Leica MS5 device): powder has 0.5–1.1 mm size, while granules have 2.5–3.7 mm size. The average density of GTR aggregates (determined by Micromeritics AccuPyc 1330 He-pycnometer, Norcross, GA, USA.) is 1202 kg/m³.

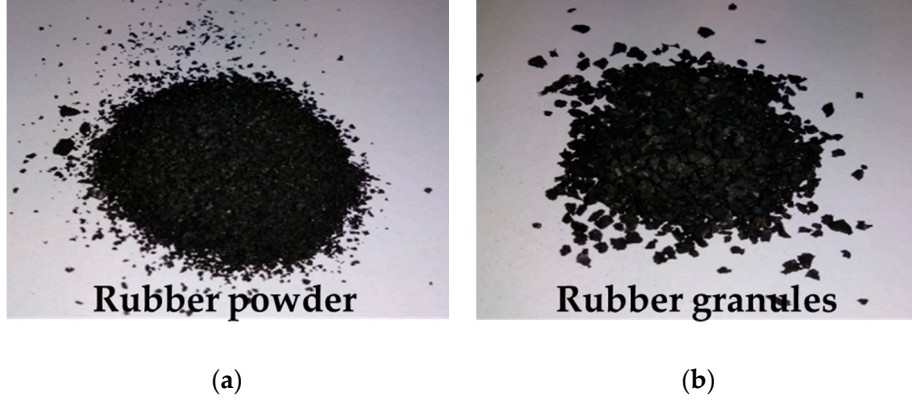

(**a**)                                         (**b**)

**Figure 7.** Size and shape of GTR powder (**a**) and granules (**b**) used in the research work.

Starting with the reference formulation, the following sand-GTR volume replacements were performed: in SSM, sand was totally replaced with rubber powder; in CSM, sand was replaced by 25 vol.% rubber powder and 75 vol.% rubber granules. As mentioned earlier, the fresh mix rheology and the raw materials balance must be optimized according to the extrusion system features (nozzle design, pumping system, printing process parameters) and proper printability requirements. To establish the appropriate mixture proportions, a qualitative investigation was performed on the flowability, extrudability, and buildability of the fresh mixtures. For this purpose, six-layer slabs with dimensions 220 mm × 160 mm × 55 mm were printed with a linear extrusion speed of 33 mm/s. Below is a description of the test procedure to evaluate the optimal mix design for each cement-based compound:

- Flowability: it defines to the ease with which the fresh mix flow out of the nozzle without obstruction. This printing criteria was evaluated by visual inspection of the deposition process.
- Extrudability: it refers to the fresh material ability to be continuously deposited through the extrusion nozzle with good dimensional conformity/consistency and without defects. Extrudability was qualified as "correct" if the printed object was completely free of discontinuity and voids.
- Buildability: it can be evaluated by the number of layer of the sample that can be printed without collapse or relevant deformation. In this study, 6 layers was adopted as the target requirement to accept the mixture as "printable".

Figure 8 shows some steps of the printability test.

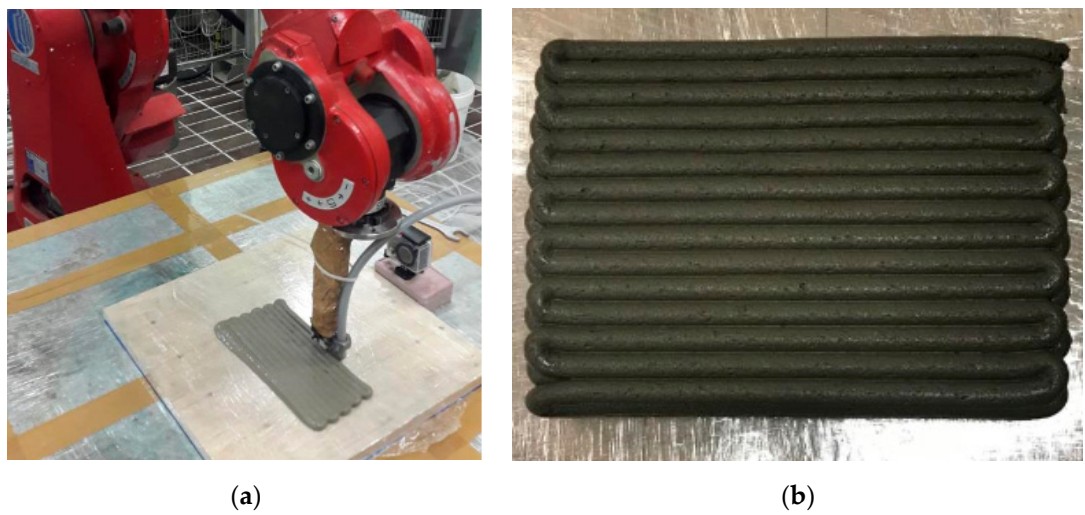

(**a**)                                         (**b**)

**Figure 8.** Printability test: slab extrusion process (**a**) and consistency of the optimum fresh mix (**b**).

For each rubberized formulation, the water content has been gradually varied in order to obtain similar printability properties as Ref. mix. Table 3 below shows the printable mix designs, referring to 1 m³, for each compound.

**Table 3.** Mix proportions of cementitious aggregates.

| Mix Design | Ref. | SSM | CSM |
|---|---|---|---|
| *w/c* ratio | 0.375 | 0.325 | 0.287 |
| Sand (kg/m³) | 1100 | 0 | 0 |
| GTR pwd. (kg/m³) | 0 | 300 | 75 |
| GTR gran. (kg/m³) | 0 | 0 | 240 |
| Additives (kg/m³) | 152 | 152 | 152 |

As shown in Table 3, the water-to-cement ratio (*w/c* ratio) is lower in the rubberized compounds than Ref. mix. Besides, this decrease is more relevant in CSM mix containing the coarse rubber filler. The aforementioned effect is related to the greater size and hydrophobic nature of the polymer particles compared to limestone sand. According to Lyse's rule [17], as the aggregate size increases, the water required to reach specific workability of the fresh material decreases. The lower use of water for the production of rubberized mortars, compared to the standard cement-based material, can be considered a beneficial aspect with regard to optimization in the use of natural resources for the construction industry.

### 3.3. Samples Manufacturing

After air-dry hardening, a preliminary observation on the printed slabs (see Figure 9) was performed to evaluate the print quality of the mixtures (presence of defects, surface finish, inter-layer adhesion). Then, 48 mm × 42 mm × 22 mm specimens (four samples for each mixture) and 70 mm × 40 mm × 40 mm specimens (two samples for each mixture) were extracted from the slabs by cutting with a diamond circular saw. The first samples were used for porosity test, optical microscopy analysis, and water contact angle test. The second ones were characterized by water sorptivity test.

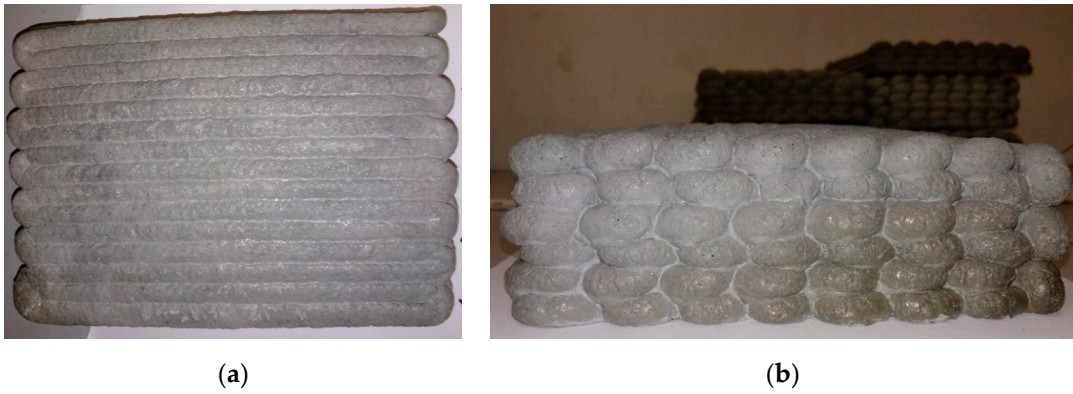

(**a**)           (**b**)

**Figure 9.** Top-view (**a**) and front-view (**b**) of a printed slab (SSM mix).

### 3.4. Porosity Measurements

The pore system of cement-based materials can adversely affect both mechanical behaviour (failure strength and creep strain) and transport properties. Transport properties are intimately related to the durability of the material: higher porosity promotes the permeation of physical-chemical agents (liquids, gasses, various aggressive ions, pollutant agents) resulting in material degradation [18].

Vacuum saturation method (ASTM C1202) [19] was used to determine the permeable porosity of cement-based mixtures. The specimens were dried in the oven at 110 °C for 2 h to determine the oven-dry mass ($W_{dry}$). Subsequently, the specimens were placed inside a desiccator connected to a

vacuum pump. The pump was started and run for 4 h at a pressure of 0.3 bar to evacuate all air from the open pores. After the process, the samples were weighed and the saturated surface-dry mass of the specimen in air ($W_{sat}$) was evaluated. Then, bidistilled water was gradually fed into the desiccator, until the blocks were fully covered. 24 h later samples were taken out of the liquid and weighed ($W_{wat}$). The permeable porosity ($P_{\%}$) was calculated based on Equation (1).

$$P_{\%} = \frac{W_{sat} - W_{dry}}{W_{sat} - W_{wat}} \tag{1}$$

Optical microscope (OM) images, acquired with Leica MS5 device, were analyzed by ImageJ software (Bethesda, DC, USA.) to evaluate the distribution of open porosity and surface defects in each sample.

*3.5. Water Contact Angles (WCAs) Test*

Water contact angles (WCAs) on the reference and rubber-cement composites surfaces were measured using an $OCA_{15}$Pro analyzer (DataPhysics Instruments, Filderstadt, Germany). Analyses were performed by the sessile drop method and using ultrapure water as testing liquid. Twelve drops with a fixed volume of 3 μL were deposited (dosing rate = 1 μL/s) and analyzed for each cement-based formulation. WCA measurements were determined ten seconds after the water drop deposition. From the twelve depositions made, an average WCA value was calculated. The study was performed to evaluate the effect of hydrophobic rubber fillers on cement matrix wettability.

As confirmed by several studies [20,21], reducing the wetting properties of concrete can be an interesting strategy to improve resistance to moisture and corrosive liquid permeation, one of the main factors of building materials deterioration. Besides, the development of a hydrophobic cement matrix can imply other interesting benefits such as icephobicity, self-cleaning behavior, and resistance to paints and graffiti [15].

*3.6. Water Sorptivity Test*

According to Elawady et al.'s definition, sorptivity is the tendency of a cement-based material to absorb and transmit water and other liquids by capillarity [22]. It was recognized as an important chemical-physical index of durability performance because the experimental method used for its evaluation reflects the way that most building materials will be penetrated by water and other corrosive agents [23]. Capillary absorption depends on two factors: (a) porous structure of the material (porosity degree, tortuosity, pore size); (b) physical-rheological properties of permeating fluid (surface tension, density, viscosity) [15].

Sorptivity index (S) was evaluated in accordance with ASTM C1585 [24]. The test is based on determining the water absorption rate by measuring the mass increase of the sample as a function of time and in the unidirectional flow condition. The samples were placed in a Pyrex glass pan on Aluminum rods. The function of the rods is to avoid contact between the test surfaces and the bottom of the pan and thus ensure correct material-liquid exposure. Bidistilled water in the pan was maintained at about 1 cm above the base of the specimens throughout the experiment. The test was performed for 40 days during which, at regular time intervals of 24 h, the mass of the samples ($W_n$) was determined. A total of 15 mass measures were recorded. In each detection, the temperature of the water (T) was also measured in order to obtain an accurate density value of the liquid ($\varrho$). Prior to the test, the sides of the specimens were coated by silicone rubber to ensure unidirectional water absorption and minimize the liquid side evaporation during the test. Dry samples were weighed ($W_i$) before starting the experiment. Figure 10 shows sorptivity specimens during the test.

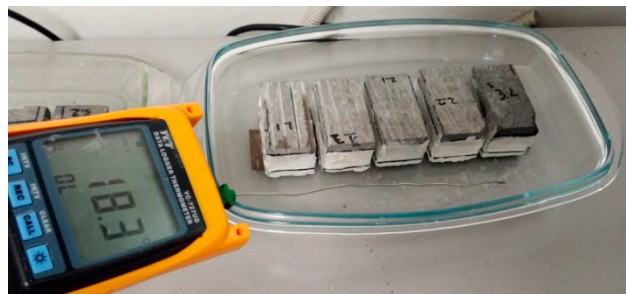

**Figure 10.** Cementitious specimens during water sorptivity test.

Sorptivity index (S) was obtained by using Equation (2):

$$\frac{(W_n - W_i)}{\rho(T) \times A} = S \times \sqrt{t} \tag{2}$$

where A = the test surface of the specimen that was in contact with water (mm$^2$); t = time (min). To determine S index, the cumulative volume of water absorbed per unit of area (1st member of the equation) was plotted against t$^{1/2}$. S (mm/min$^{0.5}$) is the slope of the best fit line to the absorption curve.

## 4. Rubberized Cementitious Aggregates: Cost-Benefit Analysis

Concrete and mortar have become very expensive materials because the high price of aggregates which constitute three-quarters of the mixture volume [25]. Incorporation of waste tire rubber fillers as mineral aggregates replacement in cement-based compounds could be beneficial not only in engineering and environmental terms but also in manufacturing costs of the aforesaid building materials. Table 4 below lists the average costs of the mineral and rubber aggregates used for the development of the three mixtures investigated in this work.

**Table 4.** Average cost (per ton.) of aggregates used in the production of cement mixtures.

| Aggregate Type | EUR/ton | Source |
| --- | --- | --- |
| Limestone sand (0–0.4 mm) | 120 | [26] |
| GTR powder (0–1 mm) | 140 | ETRA |
| GTR granules (2–4 mm) | 170 | ETRA |

The lightly higher (but not prohibitive) cost of polymer raw materials, compared to the stone aggregate commonly used as a constituent of the cement mixture, can be compensated by some interesting economic and production factors [27,28]:

- Indirect costs savings. Replacing traditional aggregates with recycled raw materials implies savings related to landfill tax, landfill gate fee and aggregates levy. The indirect costs rate depending on the socio-economic policy of the country, but an average saving of EUR 15 per m$^3$ of recycled-based cement material produced is estimated.
- Indirect benefits. Despite its initial higher unit cost, the low density of GTR fillers compared to that of limestone sand (~2600 kg/m$^3$) could also result in cost savings when the replacement is performed in terms of volume rather than weight. The use of lighter raw materials would result in reduced quantities of aggregates required. Referring to Table 3, about 300 kg of tire rubber filler instead of 1100 kg of sand is needed to obtain 1 m$^3$ of cement mixture suitable for AM. In this context, an additional savings source concerns the smaller amount of water needed for the production of rubberized mixtures.
- Tax incentives. In several countries, relevant economic incentives have been provided for companies using recycling and reuse raw materials. For example, in Italy, the "Green Economy

law" aims to financially support companies that invest in low-environmental manufacturing approaches. Under this legislation, companies can take advantage of a 10% tax credit on investments made.

## 5. Conclusions

This paper examines the usage of two types of GTR fillers (rubber powder and rubber granules) as substitutes for fine mineral aggregates in a cement-based mixture suitable for AM. The aim of the research was to investigate the synergistic effect of the different rubber aggregate grain size on material printability and certain physical-chemical properties related to performance durability. Based on the experiments performed, subsequent conclusions are reported:

1. The greater deformability of fillers compared to fine mineral aggregates implies less rigidity of the deposed filaments and therefore better inter-layer adhesion. The internal morphology of the hardened rubberized materials is homogeneous and free of structural defects, while the Ref. samples show voids and cavities due to poor layers bonding.
2. Permeable porosity of cement mixtures modified with rubber fillers is lower than the Ref. mixture. The lower $w/c$ ratio required for the realization of rubberized mixtures compared to the standard formulation minimizes the formation of pores related to the aging process. Besides, fillers synergy plays a key role in the microstructural properties of the material: rubber powder ensures the mixture compaction, while rubber granules hinder the crack propagations in the matrix.
3. The presence of rubber aggregates increases material hydrophobicity. This aspect is crucial regarding the material's inertia to moisture and damaging agents.
4. Water sorptivity test showed very good permeability performances for all the mixtures developed in this research work. In the CSM mix, the water capillary absorption is slightly higher due to the effect of interface cavities acting as conduits for liquid penetration.

Hence, the use of GTR fillers as aggregates in cement mortar has eco-sustainable credentials, in that it uses a waste product to optimize several performances and provide a material that will be potentially more durable than a plain mixture with subsequent lower life cycle costs due to reduced maintenance requirements. In addition, the use of recycled materials in the construction industry can bring significant benefits in terms of production costs and financial support.

In future research works, the experimental testing on the material will be completed (static and dynamic mechanical characterization, thermal and acoustic testing). Subsequently, according to the performances determined by the experimental characterization, the most suitable type of building and architectural application will be evaluated. The high freedom of design offered by AM can be used to perform topology optimization studies in order to improve technological properties by operating on prototype shapes and geometries.

**Author Contributions:** Experiments were designed by M.V. and M.S.; M.S. and D.M. carried out the experiments, which were also supervised by M.V.; M.S. written and D.M. and M.V. contributed to the discussion of the results and to the revision of paper. All authors have read and agreed to the published version of the manuscript.

**Funding:** This research was also performed thanks to "Sapienza" University direct financing, for PhD student Matteo Sambucci, called "Avvio alla Ricerca".

**Acknowledgments:** The authors express their sincere gratitude to Valeria Corinaldesi and Glauco Merlonetti (Marche Polytechnic University) for their support in the additive manufacturing process. Besides, the authors would like to thank MariaGabriella Santonicola and Elisa Toto (Sapienza University of Rome) for their technical assistance in water contact angle measurement.

**Conflicts of Interest:** The authors declare no conflict of interest.

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
