# Peer review of "Tire Recycled Rubber for More Eco-Sustainable Advanced Cementitious Aggregate"

_recycling, doi:10.3390/recycling5020011_

Round 1

Reviewer 1 Report

The paper presents very interesting findings.

I have a few minor and one major comment regarding this paper, please find them below:

  • The powders produced by tire grinding are commonly called - ground tire rubber (GTR). It is well established description in the literature, please use it.
  • What is the "w/c ratio"? Please explain in the text.
  • The authors explain formation of the interlayer voids for the Ref sample by poorer adhesion between the filaments. However, from the physico-chemical point of view the adhesion for the Ref sample sould be higher due to higher surface enegry and polar character of the sand in contrast to non-polar rubber surface. In my opinion the formation of inter-layer voids between the Ref filaments stemms from higher surface tension of the extruded Ref mixture (see the lower contact angle values). The non-polar character of the rubber additives decreases the overall surface tnsion of the mixes resulting in easier flow that allows filling the gaps. Also, the authors relate this effect to elastomeric properties of rubber, which is in my opinion not true - vulcanized rubber is not able to flow due to its cross-linked chemical structure. What we see here is the effect of lower interfacial interactions between the rubber particles and the cement additives, that result in lower surface tension of the mixes, thus easier flow and filling the gaps before hardening.
  • Be consistent for the contact angle measurement abbreviation - WCA or OCA?
  • The pictures showed in Figure 7 b & c are discussable because the tested surface seems to be non-flat - especially in Figure 7 c the water droplet seems to be located on some surface defect.

Author Response

In attached the cover letter with the answere

Reviewer 2 Report

Manuscript ID: recycling-777215

Title: Tire Recycled Rubber for More Eco-Sustainable Advanced Cementitious Aggregate

The manuscript reports that preparation and chemical-physical performances of the cement hybrids by using tire recycled rubber fillers with different grain sizes. The present experiments are not novel in this manuscript. However, I cannot deduce the novelty of this paper. Therefore, I don’t recommend it for publication.

Author Response

(The authors gave the same response as above.)

Reviewer 3 Report

"Tire Recycled Rubber for More Eco-Sustainable Advanced Cementitious Aggregate" is very well written manuscript describing research focused on using tire rubber fillers with different grain sizes as a replacement for the mineral aggregates used in a cement-based mixture suitable for extrusion-based additive manufacturing.

Although the authors state: „Rubber fillers do not alter the rheological properties of the cement material, which was properly extruded with better print quality than the reference mixture." it is not clear how the rheology was evaluated. Rheological properties of cement-based mixtures should be commented - how did they decide which different ratios of water to use?

Please correct all Kg into kg.

In my opinion the Significance of Content is average, if you had included the mechanical properties in the manuscript it would be rated high.

Author Response

(The authors gave the same response as above.)

Round 2

Reviewer 2 Report

Manuscript ID: recycling-777215

The manuscript was revised carefully and improved so much according to reviewers’ suggestions. The scientific insights are expressed well in this manuscript. Overall, the current revision is recommended for publication in the Recycling.